# Gibberellin Signaling Repressor *LlDELLA1* Controls the Flower and Pod Development of Yellow Lupine (*Lupinus luteus* L.)

**DOI:** 10.3390/ijms21051815

**Published:** 2020-03-06

**Authors:** Katarzyna Marciniak, Krzysztof Przedniczek

**Affiliations:** Department of Plant Physiology and Biotechnology, Faculty of Biological and Veterinary Sciences, Nicolaus Copernicus University, Lwowska 1 Street, 87-100 Toruń, Poland; k.przed@doktorant.umk.pl

**Keywords:** DELLA, gibberellins, flowers, pods, yellow lupine

## Abstract

Precise control of generative organ development is of great importance for the productivity of crop plants, including legumes. Gibberellins (GAs) play a key role in the regulation of flowering, and fruit setting and development. The major repressors of GA signaling are DELLA proteins. In this paper, the full-length cDNA of *LlDELLA1* gene in yellow lupine (*Lupinus luteus* L.) was identified. Nuclear-located LlDELLA1 was clustered in a second phylogenetic group. Further analyses revealed the presence of all conserved motifs and domains required for the GA-dependent interaction with Gibberellin Insensitive Dwarf1 (GID1) receptor, and involved in the repression function of LlDELLA1. Studies on expression profiles have shown that fluctuating *LlDELLA1* transcript level favors proper flower and pod development. Accumulation of *LlDELLA1* mRNA slightly decreases from the flower bud stage to anther opening (dehiscence), while there is rapid increase during pollination, fertilization, as well as pod setting and early development. *LlDELLA1* expression is downregulated during late pod development. The linkage of *LlDELLA1* activity with cellular and tissue localization of gibberellic acid (GA_3_) offers a broader insight into the functioning of the GA pathway, dependent on the organ and developmental stage. Our analyses provide information that may be valuable in improving the agronomic properties of yellow lupine.

## 1. Introduction

Interest in the cultivation of legumes, including lupine (*Lupinus* L.), is growing significantly. An important feature of these species is the ability to bind atmospheric nitrogen, which significantly enriches the soil and reduces energy costs to manufacture fertilizers. The high fodder value of lupine, universal consumption values, and its role in sustainable and ecological production are being widely accepted and appreciated [1]. Increased consumer awareness about the health benefits of legumes, mainly the proteins occurring in seeds, stimulates their production [2,3]. Major lupine producers are Australia, Poland, Russia, Germany, Belarus, and Chile [4]. Nevertheless, a key problem in growing lupine in rapidly changing environmental conditions is premature and excessive generative organ abortion, which significantly reduces the yield. This effectively discourages the cultivation of this species by innovative farmers who expect large and stable yields every year. Therefore, it is extremely important to understand the basic mechanisms responsible for the formation, growth, and development of lupine flowers and pods at the genetic, molecular, and hormonal levels. 

DELLAs are proteins that are able to integrate multiple environmental and endogenous signals to control various aspects of plant growth and development, including flowering, and pod setting and development [5,6,7]. Physiological and molecular investigations have shown that DELLAs, considered as a major negative regulator of gibberellin (GA) signaling, connect almost all phytohormonal pathways [7]. These proteins inhibit the DNA-binding capacity of transcription factors (TFs) or the activity of transcriptional regulators (TRs) from different families [8]. DELLAs represent a subset of the plant-specific GRAS (GAI—GA Insensitive, RGA—Repressor of GA1-3, and SCR—Scarecrow) family of TRs [9]. Common to the GRAS proteins, DELLAs have a conserved C-terminal GRAS functional domain containing: (I) putative nuclear localization sequence (NLS); (II) leucine heptad repeats (LHR1 and LHR2), which mediate protein–protein interactions; and (III) conserved motifs—VHIID, PFYRE and SAW, which enable secondary interactions with the GA receptor Gibberellin Insensitive Dwarf1 (GID1) and F-box proteins (FBPs) [9,10]. In contrast to other GRAS proteins, DELLAs have three specific domains located at the N terminus: (I) the DELLA, and (II) the TVHYNP domains, both responsible for the interaction between DELLAs and GID1, as well as (III) polymeric Ser/Thr/Val motifs (poly S/T/V), which could be targets of phosphorylation or glycosylation [7,11,12,13]. 

*DELLA* genes have been identified in several plants, e.g., tomato (*Solanum lycopersicum PROCERA*), grapevine (*Vitis vinifera GAI1*, *GA-INSENSITIVE1*), and among cereals: rice (*Oryza sativa SLR1*, *SLENDER RICE1*), barley (*Hordeum vulgare SLN1*, *SLENDER1*), wheat (*Triticum aestivum RHT-1*, *REDUCET HEIGHT-1*), and maize (*Zea mays D8/9*, *DWARF8/9*) [14,15,16,17,18,19]. In these species, the single-copy, highly conserved *DELLA* gene was recognized. In many other plants, the *DELLA* has undergone amplification. In the model plant *Arabidopsis thaliana,* five *DELLA* genes can be distinguished: *GAI*, *RGA* (*REPRESSOR OF GA1-3*), *RGL1* (*RGA-LIKE1*), *RGL2,* and *RGL3,* which exhibit partial functional redundancy [20,21,22,23,24]. Thus, *GAI* and *RGA* control cell expansion and division in stem and root, as well as floral induction [25,26]; *RGL2* is the key inhibitor of seed germination [23]; *RGA*, *RGL1*, and *RGL2* control floral development [27,28]; and *RGL3* contributes to plant fitness during environmental stress [24]. The distinct *DELLA* functions depend on promoter-specific regulation and, consequently, highly tissue-specific gene expression [7]. Multi-copy *DELLA* gene was also recognized in *Lactuca sativa*, *Populus trichocarpa*, *Malus domestica*, *Artemisia annua*, *Prunus salicina*, and *Prunus mume*, including legumes such as *Glycine max*, *Phaseolus vulgaris*, *Medicago truncatula*, and *Pisum sativum* [29,30,31,32,33,34]. The degradation of DELLAs induced by GAs, with the participation of the GID1 and FBPs (SLY, SLEEPY; SNE, SNEEZY in *A. thaliana* or GID2 in *O. sativa*), mediates a key event in GA signaling pathway. In the absence of phytohormone molecules, DELLAs repress GA-mediated responses in plants. When GAs are present, they bind to receptors and cause further interaction between GID1 and the DELLAs. With the aid of the SKP, CULLIN, F-box (SCF) complex, DELLAs are degraded via the 26S proteasome pathway [35,36,37]. 

GAs are phytohormones that are essential for many processes in the plant life cycle. The involvement of these bioactive molecules in the flowering process (floral induction, evocation, and morphogenesis) is beyond doubt, clearly demonstrated in a study on GA metabolism and signaling mutants [5,38,39]. Further stages of generative development, such as fruit/pod setting, depend on successful anther dehiscence with the release of viable pollen grains, pollination, and fertilization, which trigger the appropriate developmental program through the activation of hormonal signaling pathways [40]. In this regard, fruit/pod development is a multiphase process that requires a tight coordination of molecular, biochemical, and structural elements. The series of modifications that control the transition of fruit/pod growth through consequent developmental stages involve many distinctive pathways [34]. In the present study, the main aim was to verify that *LlDELLA1*, considered as the repressor of the GA signaling pathway, is involved in the proper generative organ development in yellow lupine (*L. luteus* L.), crucial for maintaining high yields. In total, 18 phases of flower and pod development were selected and the expression profile of *LlDELLA1* was examined. Histological analyzes revealed the structure of anthers and pods, which made it possible to determine the early and late stages of generative organ development in yellow lupine. On this basis, the moment of dehiscence with the release of pollen grains, pollination, and fertilization was also established. Further immunohistochemical analyzes, with the use of primary and secondary antibodies, have allowed for cell and tissue localization of gibberellic acid (GA_3_) in selected stages of pod development. The combined study of physiological processes at the cellular and molecular level appears to provide a full and unique insight into the important aspects of plant growth and development. The aim of this paper is to characterize the reproductive processes in crop species in order to develop effective breeding systems to improve yields.

## 2. Results

### 2.1. Cloning and Analysis of Full-Length cDNA Encoding L. luteus DELLA1 Protein

The full-length cDNA of *LlDELLA1* gene (2067 bp, Appendix A) was identified by using degenerate primers (561 bp; Appendix A), 3′ RACE-PCR technique (535 bp; Appendix A) and sequence derived from RNA-Seq experiment (1013 bp). CDS is 1677 bp in length. LlDELLA1 was predicted to encode protein of 558 aa residues with calculated molecular weight of 61.321 kD and isoelectric point = 5.37 (Appendix A). For in silico analyzes, 23 amino acid sequences exhibiting high similarity/identity and derived from model species, cereals, and others closely related to *L. luteus* were typed using BlastP. To determine the relationship within selected DELLAs, a phylogenetic tree was constructed (Figure 1). This tree, which included three groups (I–III), revealed that DELLAs originate from a common ancestor. Group I was the monocotyledonous group that contained two DELLAs. Groups II and III were dicotyledonous groups. LlDELLA1 was clustered into the second group with other GAI and RGA proteins. Two proteins from *Glycine max* (GmDELLA4 and GmDELLA6) were not assigned to any group, which indicates their distant origin. It should be noted that there is also a Group IV that includes DELLA-like proteins from moss and gymnosperm species (not included). The phylogenetic tree also showed that LlDELLA1 and GAIP-B-1 from *L. angustifolius* form a closely related subgroup. Additionally, these lupine proteins share similarity with tree GAIP-B proteins from *Arachis duranensis, A. ipaensis,* and *A. hypogaea*. Additionally, analysis using BlastP showed that *L. angustifolius* GAIP-B-1 has 93.38% identity to LlDELLA1, while GAIP-B from *A. duranensis*, *A. ipaensis*, and *A. hypogaea* have 73.31%, 73.11%, and 72.49% identity with LlDELLA1, respectively. In order to demonstrate the relationship between DELLAs occurring in *L. luteus* and *A. thaliana*, Appendix A was prepared. The similarity was determined in relation to the maximum agreement of amino acid sequences for AtGAI-AtRGA pair of proteins and a value of one was assumed in relation to this agreement (81% identical amino acids). Interestingly, all other AtDELLAs show only 56% to 71% amino acid sequence identity. The level of 68% or 65% sequence identity confirms that LlDELLA1 corresponds to GAI/RGA (Appendix A). Additionally, LlDELLA1 has nuclear localization like all DELLAs derived from *A. thaliana* (Appendix A).

To better understand the probable function of LlDELLA1, all characteristic, typical, and conserved motifs were discovered and their locations determined against the background of other DELLAs in various plant species (Figure 2; Figure 3).
All selected proteins with 503 to 635 aa show a very similar arrangement of motifs, with small deletions or inserts (Figure 2). As expected, LlDELLA1 contains at the N terminus a DELLA regulatory domain with DELLA and TVHYNP motifs (Figure 2; Figure 3; Appendix A). All highlighted amino acid residues in the DELLA motif containing DeLLaΦ**L**xYxV sequence, LExLE motif with the consensus sequence MAxVAxxLExLExΦ, and in the TVHYNP motif (TVhynPxx**L**xxWxxxMxx) are essential for the direct interaction between the DELLA and GID1 receptor surface (‘Φ’ represents a non-polar residue, and ‘x’ represents any residue) [34]. In the case of LlDELLA1, as well as other DELLAs from the closest related species, the substitution of L into V (DeLLaΦ**L**xYxV—DeLLaΦ**V**xYxV) and L into I (TVhynPxx**L**xxWxxxMxx—TVhynPxx**I**xxWxxxMxx) took place. In common **L**EQLE motif, the first Leu is replaced with Ile in *L. luteus* and *L. angustifolius* (Appendix A). At the C terminus of LlDELLA1, the GRAS functional domain with RKVATYFAEALARR (nuclear localization signal, NLS), VHVID, RVER, and SAW motifs were found (Figure 2; Figure 3; Appendix A/C). These motives are strictly conserved, implying their import role in biological function. All these motifs are part of DELLA, TVHYNP, POLY S/T/V, LHR1/2, VHIID, PFYRE, and SAW domains (Figure 3; Appendix A). 

The tertiary structure of LlDELLA1 protein was predicted using different tools (Figure 4). The model calculated by Robetta has 17 α-helixes, β-sheet with 9 strands, and covers the entire amino acid sequence (Figure 4A). The model has 63% confidence level, which might be due to the lack of an appropriate template from Protein Data Bank (PDB) in first ~180 aa (Figure 4C). The phyre^2^ was able to predict a 3D structure in part of LlDELLA1 from 186 aa up to 554 aa (Figure 4B). This model has 11 α-helixes, β-sheet with 9 strands, and 100% confidence; it covers 66% of the sequence. The best templates used to calculate LlDELLA1 3-D model were SHORT-ROOT (SHR) from *A. thaliana* (PDB: 5B3H), SCARECROW-LIKE PROTEIN 7 (SCL7) from *O. sativa* (PDB: 5HYZ), SCARECROW from *A. thaliana* (PDB: c5b3hD), and GAI from *A. thaliana* (PDB: c2zshB). The AtGAI has the most identical sequence from all PDB templates to the LlDELLA1 sequence (Appendix A). Additionally, a secondary structure of LlDELLA1 was constructed (Appendix A).

### 2.2. General Expression Profile of LlDELLA1 in Various Organs of L. luteus

The analysis of the expression profile of *LlDELLA1* in vegetative (leaf blades, petioles, stipules, stem, roots) and generative (flowers, pods) tissues of yellow lupine during generative development—fully developed flowers (FDF) and fully developed pods (FDP)—showed the highest mRNA level in flowers, and subsequent ~3 times less in leaf blades during the FDF phase, as well as in pods and leaf blades during the FDP phase (Figure 5). In other examined organs, regardless of the developmental phase, lower and relatively constant transcriptional activity of *LlDELLA1* was noticed. 

### 2.3. Histological Analyses

Histological analyzes were performed in order to precisely determine the individual developmental stages of generative organs. Ten stages of flower development (1F–10F) were selected (Figure 6A). Previous studies on *L. angustifolius* and *L. albus* [41,42] have shown that pollination and fertilization occur at very early stages of flower development. Therefore, it was decided that the phase of flower development in which anther opening takes place is indicated, which will correlate with the time of pollination followed by fertilization. It was established that the *L. luteus* anthers open in the fourth stage of flower development (4F), when they are completely closed (Figure 6B). Thus, *L. luteus* is a highly self-pollinating species. 

In the next stages of flower development, pollination and fertilization occurred, and pods were set up. The yellow lupine pod is a dry fruit developed from a single pistil (Figure 6C). The pericarp (pod wall) encloses and protects the developing seeds, contains photosynthetic machinery to sustain itself and the seeds, and accumulates nutrients as a sink tissue that are later remobilized during seed development. The early stages of pod development were established (Figure 6C) up to the 10th phase (5F–10F; Figure 6A). A very large number of small dividing cells with cell nuclei were then identified. In the immediate post-fertilization phase, the seeds occupy most of the space between the pod walls, and then septa begin to form between seeds, enclosing them in separate chambers. The next eight phases (1P–8P; Figure 6A) are the late stages of pod development (Figure 6D). In these phases, clearly differentiated cells were observed. The cells of the outer mesocarp were much larger than adjacent cells of the inner mesocarp, where conductive bundles are present. As the pod approaches maximum dry weight, the seeds proportionally fill more space until the pod walls are pushed apart and the septa are broken. As the seeds develop, the endosperm is progressively depleted.

### 2.4. Expression Profile of LlDELLA1 during Generative Organ Development

Due to the fact that *LlDELLA1* exhibited the highest expression in generative organs of yellow lupine, it was decided to separate the subsequent stages of flower and pod development. On this basis, it was established that from the early stages of flower bud development (1F) to the moment of anther opening (dehiscence, 4F), the level of *LlDELLA1* transcripts slightly decreases, reaching the lowest level in about the 4^th^ stage of flower development (Figure 7). Between Phase 6 and 10 (6F–10F), there is a rapid increase in the amount of *LlDELLA1* mRNA, i.e., at the time when pollination, fertilization, and formation of pods and seeds take place, up to their early development, with the maximum peak in the 10^th^ phase (10F). In the later stages of pod development (1P–8P), the *LlDELLA1* expression decreases, even below the level observed during flower development (Figure 7).

### 2.5. Expression Profile of LlDELLA1 in Response to Phytohormone and Inhibitor Treatment

*DELLAs* encode proteins which integrate signaling pathways of different phytohormones. Moreover, a range of studies have shown that the level of *DELLA* transcripts changes under the influence of various phytohormones and their inhibitors [7,32,43]. In this paper, the *LlDELLA1* expression patterns after GA_3_, GA biosynthesis inhibitor—chlorocholine chloride (CCC), methyl jasmonate (MeJA), indole-3-acetic acid (IAA), and 6-benzylaminopurine (BAP) treatments were analyzed (Figure 8). The obtained results showed that *LlDELLA1* has a similar expression in response to MeJA, IAA, and BAP applications, comparing to the control variant. Statistically significant differences in *LlDELLA1* transcriptional activity was noticed after GA_3_ and CCC treatment. *LlDELLA1* transcripts were induced in response to the CCC application. In contrast, in response to the GA_3_ treatment, *LlDELLA1* mRNA level was suppressed.

### 2.6. Immunolocalization of GA_3_

In order to understand whether there was a correlation between the transcriptional activity of the *LlDELLA1* and the level of GA_3_, immunohistochemical analyses were performed. In the early phase of pod development (10F), moderately low but noticeable GA_3_ level was observed in the pod wall (Figure 9C,C’). The fluorescence signal indicating the presence of GA_3_ was mainly dispersed in the whole cytoplasm, while less visible near the cell walls (Figure 9C’). In turn, in 1 mm seeds, the phytohormone molecules were undetectable (Figure 9A,A’,B,B’). Additionally, a huge number of cell nuclei, as well as dividing cells, were found in young and mature pods, which proves the rapid growth and development of these kinds of cells. On the other hand, a higher level of GA_3_ was detected in the 5 mm seeds (Figure 10C,D,E) than in the pod walls (Figure 10A,B) during the late phase of pod development (8P). In this case, the green fluorescence signal was mainly observed near the walls of the seed cells (Figure 10C,D) and only in some cells were the signal distributed throughout the cytoplasm (Figure 10E). In the case of pericarp, the presence of GA_3_ was not visible for epidermis, outer, and inner mesocarp cells, as well as conductive bundles (Figure 10A,B).

## 3. Discussion

Gibberellins play a crucial role in the development of generative organs [38]. The DELLA proteins are negative regulators of GA signaling in many plant species, including *A. thaliana*, *O. sativa*, *H. vulgare V. vinifera,* and *Gossypium hirsutum* [16,17,20,21,22,23,24,44,45]. In crop legumes, including lupines, these issues have not yet been well understood. Moreover, excessive and premature flower and pod shedding in yellow lupine is an economic disadvantage, as proper formation and development of generative organs is essential for plant productivity. In order to be able to control this process, a wide-ranging fundamental knowledge of the molecular and hormonal mechanisms of flower and pod development is required.

### 3.1. LlDELLA1 Encodes Protein Containing Characteristic and Conserved Motifs and Domains 

The complete cDNA of the *LlDELLA1* gene was isolated and sequenced from yellow lupine. The deduced amino acid sequence of LlDELLA1 displays structurally high similarity with other DELLAs, especially from the Fabaceae family, and comprises of the two domains essential for protein function, including the N-terminal DELLA domain (DELLA and TVHYNP motifs) [46] and the highly conserved C-terminal GRAS domain [9]. Both domains are required for GA-dependent interaction with the GID1 receptor and are involved in the repression function of the protein [10,34,43,47]. 

Our analyses have shown that in the DELLA motif, usually containing the DeLLaΦ**L**xYxV sequence [34,48], the third Leu residue is substituted by the distinct amino acid Val, not only in LlDELLA1 from *L. luteus*, but also from other closely related species. In the following **L**ExLE motif, with the consensus sequence MAxVAxx**L**ExLExΦ, the first Leu is replaced with Ile, similarly as in the TVHYNP motif (TVhynPxx**L**xxWxxxMxx). These alterations in amino acids are unlikely to affect direct interaction between the DELLA and GID1 receptor; however, this requires further investigations. Lu et al. [33] have shown that in *Prunus mume*, the PmDELLAs are most divergent in their N-termini and highly homologous over their C-termini. These findings are also consistent with previous studies, including apple (*Malus domestica*) [49,50]. In turn, it is well known that serve deletions of whole DELLA motif in *A. thaliana* convert proteins into a GA-unresponsive, constitutively active GA signaling repressor [51]. In LlDELLA1, the poly S/T/V motif was also recognized, where in *A. thaliana,* this kind of motif is a possible site for phosphorylation or glycosylation. It has been reported that the degradation of AtDELLAs first requires dephosphorylation [31]. In *A. thaliana*, the poly S/T/V motif contains the L(K/R)XI motif likely involved in binding an undetermined GA signaling component [52].

The GRAS domain is involved in the regulation of many developmental processes [49]. The motifs—RKVATYFAEALARR (NLS), VHIID, RVER, and SAW are strictly conserved in the LlDELLA1, implying their import role in biological function. In *O. sativa* or *H. vulgare*, mutations in the GRAS domain result in loss of DELLA repressor function, leading to a tall or slender plant growth phenotype [16,17,46]. GRAS domain proteins are a large family of TRs unique to plants, and conserved, e.g., in mosses, *O. sativa* and *A. thaliana* [53]. Thus far, only one GRAS protein has been demonstrated to directly bind to DNA, a legume protein called MtNSP1 (*M. truncatula* Nodulation Signaling Pathway1), suggesting that most GRAS proteins may indirectly regulate gene transcription [54]. Summarizing the knowledge gained so far about DELLAs, it has been proposed that these proteins function: (1) as coactivators of genes that negatively regulate GA signaling, (2) as repressors of transcriptional activators by blocking the ability of TF to bind its promoter, and (3) as factors that recruit chromatin remodeling complexes to promoter elements [52]. In this paper, the obtained 3-D models predict the occurrence of tertiary features similar to AtGAI and proteins from the GRAS family. The use of both methods allowed the creation of models that have two distant domains, similar to functional DELLAs from other species. The high sequence similarity, as well as the similarity of the tertiary structure suggests that LlDELLA1 may have a function parallel to other DELLAs.

### 3.2. Nuclear Protein LlDELLA1 is Clustered within Group II with GAI and RGA

Generally, DELLAs as a group of archaic proteins expand after divergence of the plants from protest and fungi; these proteins may have significant roles in the evolution of plants. This is consistent with the evolution of GRAS proteins [33]. The phylogenetic relationship between DELLAs from different angiosperm species reveals the presents of a monocotyledonous Group I and dicotyledonous Groups II and III. This indicates that DELLAs were diversified after the monocot-dicot split. However, there is a little information on the evolution of DELLAs in moss and gymnosperms species. Our analyses indicate that LlDELLA1 is clustered within the second group with GAI and RGA proteins derived from different plant species.

The bioinformatics predictions of protein localization in cells indicate a nuclear position of LlDELLA1. This result is consistent with the function of TR as expected, which should control the expressions of downstream genes. In *A. thaliana*, the AtGAI contains two basic regions that are characteristic of NLSs. The first region (206 RKVATYFAEALARRIYR 222) exactly fits the consensus for bipartite NLSs, which has been defined as two basic amino acid residues, a spacer region of ~10 residues, and at least three basic residues out of the next five. In addition, GAI contains a second basic region (134 KRLK 137) that conforms to the consensus (K-R/K-XR/K) proposed for non-typical SV40-like NLSs [20]. The presence of these sequences suggests that different GAI proteins may be targeted to the nucleus. Interestingly, GAI also contains two motifs, 169 VHALL 173 and 370 LHKLL 374, which are, respectively, closely related and identical to a consensus motif (LXXLL) that has been shown to mediate binding of transcriptional coactivators to nuclear receptors [20].

### 3.3. Generative Organs Contain the Highest Level of LlDELLA1 Transcripts

Many molecular studies have shown that the expression of *DELLA* genes differ among organs at various developmental stages. For example, *AtRGA* is expressed ubiquitously in most tissues (including seedlings, roots, rosette leaves, whole rosette plants, bolting stems, mature stems, flower buds, young siliques and mature siliques), whereas *AtGAI* is moderately transcriptionally active, with the highest mRNA levels in imbibed seeds and the lowest transcripts levels in seedlings, roots, rosette leaves, and siliques. In turn, *RGL1*, *RGL2*, and *RGL3* are expressed at higher levels in germinating seeds, young seedlings, and/or flowers and siliques, but produce low amounts of transcripts in most vegetative tissues [28,34,55]. In this study, the *LlDELLA1* expression profile was assessed during various generative developmental stages (FDF, FDP) in all organs to provide further credence about their role in regulating *L. luteus* growth and development. Preliminary analysis of the *LlDELLA1* expression showed the highest transcript accumulation in mature flowers and pods, which indicates commitment in regulation of their development. Generally known, that DELLAs are required in floral development, as well as in seed dormancy or germination [28,56]. In *L. luteus* vegetative tissues, regardless of the developmental phase, basal mRNA level of *LlDELLA1* was observed, which suggests that suppression of its expression may be required for proper development; however, the moderate accumulation level of *LlDELLA1* is necessary for controlled cell elongation and expansion in different vegetative organs, as studies on *A. thaliana* have shown [25,26,57]. In *P. mume*, *PmDELLA1* and *PmDELLA2* genes were expressed in all vegetative and generative organs, but the highest expression levels were observed in the seed in the case of *PmDELLA1*, and in the stem in the case of *PmDELLA2*. The different expression profiles of *PmDELLAs* indicated that they may have similar functions in some organs and specific functions in others [33]. Summarizing, the accumulation of *LlDELLA1* transcripts appears to be organ- and developmental stage-dependent. This prelude analysis indicated that GA-negative signaling component might be transcriptionally regulated, as suggested for its orthologues in *A. thaliana* [28] or *P. mume* [33].

### 3.4. Self-Pollination Occurs when Flowers are Closed Followed by Pod Setting and Development

Since the highest expression of *LlDELLA1* was found in generative organs, a total of 18 flower and pod developmental phases were selected. Histological analyzes became necessary, which made it possible to identify the moment of anther dehiscence, pollination, fertilization, pod and seed setting, as well as their early and late development. In many species, late anther/stamen development is defined as a stage that occurs after the opening of the flower [58,59]; however, in various lupines there is a phenomenon called kleistogamy, in which dehiscence occurs before flower opening, leading to self-pollination. Under natural conditions, most annual lupines are self-compatible and mainly reproduce by self-pollination. For example, *L. angustifolius* is almost exclusively self-pollinated [4,42]. In contrast, perennial lupine species reproduce mainly through cross-pollination due to self-incompatibility [60,61]. Many literature data have shown that fertilization in self-pollinated species (*L. angustifolius*, *L. albus*) occurs in closed flowers in the very early phases of their development [41,42]. It should be noted, however, that no species has been found to be strictly self-pollinated. In the case of *L. angustifolius*, the outcrossing rate has been shown to be low, but may vary depending on a number of factors. For *L. albus*, although pollination also occurs in very early phases of flower development, it has an outcrossing rate around 10% [4,62]. 

### 3.5. Fluctuating LlDELLA1 Expression Profile Ensures Proper Flower and Pod Development

Results obtained in this paper clearly indicate that the identified *LlDELLA1* gene is associated with the proper growth and development of generative organs in *L. luteus*. Changing and fluctuating expression profile during the overall development of flower buds, flowers and fruits/pods is different in various species. In individual stages of early flower development, the moderate amount of *LlDELLA1* transcripts favors the development of fertile flowers. The results of research conducted on *Prunus salicina* have shown that *PslGAI*, *PslRGA,* and *PslRGL* were abundantly expressed in flower buds, but showed a distinct accumulation pattern afterward [34]. As in the case of *LlDELLA1*, the level of *PslRGL* transcripts decreased in the early stages of flower bud development, while contrary to *LlDELLA1* expression, it gradually decreased during embryo development and fruit initiation. The content of *PslGAI* and *PslRGA* mRNAs steadily increased along with flower development, peaking soon after fertilization. Subsequently, both transcripts behaved similar to that of *PslRGL* mRNA by decreasing to their low levels at the end of fruit-set [34]. This is slight reversal of the situation that was observed in *L. luteus*, where during pollination, fertilization and early pod setting, the level of *LlDELLA1* began to quickly increase. Interesting studies were carried out on *S. lycopersicum*, where it was shown that the depletion of *SlDELLA* was sufficient to overcome the growth arrest normally imposed on the ovary at anthesis, resulting in parthenocarpic fruits in the absence of pollination. Parthenocarpy caused by *SlDELLA* depletion is facultative, as hand pollination restored wild-type fruit phenotype [14]. Moreover, from the point of view of our work, a significant increase in *LlDELLA1* expression during pollination and fertilization is most likely associated with proper seed formation, avoiding the development of seedless pods. It is well known that pod/fruit set is the commitment of the ovary to proceed with pod/fruit development, and is controlled by positive growth signals generated during fertilization [63]. In this way, it is ensured that the maternal structures of the ovary will escort the embryo during its development. Gibberellins and also auxins are key controllers of fruit set and early development, and are widely known for their ability to promote fertilization-independent fruit development in several species [14]. 

In later stages of fruit development, decreasing accumulation of *PslGAI*, *PslRGA,* and *PslRGL* transcripts [34] correlated with *LlDELLA1* mRNA content. Throughout fruit development, the series of modifications that make the fruit proceed through the consequent developmental stages involve many different pathways. In fruit/pod development, GAs are needed to organize cell division and expansion [64]. This suggests that down-regulation of *LlDELLA1* was associated with accelerated cell division and expansion events, resulting in visible enlargement in pod size. The effect of exogenous GA treatment in increasing fruit size and weight has been confirmed many times in several species [34,64,65].

The investigations performed by Shen et al. [32] have shown that the expression of *AaDELLA1*, *AaDELLA2,* and *AaDELLA3* in *A. annua* is suppressed in all vegetative and generative tissues (roots, stems, leaves, green young alabastrums, mature alabastrums, and flowers), except in seeds. This suggests that expression of *AaDELLA*s is required for the inhibition of GA response in dormant seeds, while suppression of *AtDELLA*s is necessary for GA response and plant development [32]. In *P. mume*, the *PmDELLA1* and *PmDELLA2* are actively transcribed in seeds and during flower and fruit developmental stages, although their expression patterns are different [33]. *PmDELLA1* showed very high expression in the seed; whereas, *PmDELLA2* showed a low expression level. In addition, these two genes showed the same expressions patterns during floral blooming. They were both down-regulated from flowers at Stage A (corresponds to the bud) to Stage B (full-bloom stage) and slightly up-regulated at Stage C (end-bloom stage), which suggests their similar functions during floral blooming [33]. By contrast, these two genes showed different patterns during fruit development. The expression level of *PmDELLA2* gradually increased during fruit development, whereas *PmDELLA1* expression decreased stage by stage [33], which is consistent with the *LlDELLA1* expression pattern. Summing up, numerous studies have reported that *DELLA* genes are involved in diverse biological processes, particularly flowering and fruit/pod setting and development, but in various plants, the GA-dependent mechanisms regulating these processes may be different, and depend on many internal and external factors [66,67,68,69]. 

### 3.6. In Yellow Lupine, LlDELLA1 is Considered as a GA Signaling Pathway Repressor

Genetic screens in *A. thaliana* and *O. sativa* have led to the identification of DELLAs as the main component of the GA signaling pathway [8,43,70]. The model of GA action relies on the observation that exogenous GA treatments were associated with DELLA degradation to rescue dwarfism of a GA-deficient mutant [7,71]. *DELLA* genes are defined as repressors of GA signaling, due to the dwarfism observed in the gain-of-function mutants, whereas a slender or tall phenotype characterizes the loss-of-function mutants. GAs also regulate the transcriptional activity of *DELLA* genes [16,17,20,55]. In this paper, we revealed that *LlDELLA1* expression is suppressed in response to GA_3_ treatment. Furthermore, application of GA biosynthesis inhibitor caused an increase in *LlDELLA1* mRNA level. It is very likely that *LlDELLA1* is also a negative regulator of GA signaling pathway in *L. luteus*. The suppression of *LlDELLA1* expression by GA_3_ treatment is similar to that observed in different plant species, e.g., *BnSLY1* in *Brassica napus* [43] or *AaDELLA* in *A. annua* [32]. We also revealed that the amount of *LlDELLA1* transcripts does not change after the application of JAs, IAA, or CKs.

### 3.7. Accumulation of GA_3_ in Pericarp and Seed Depending on the Developmental Phase of Pods

The previous studies conducted on *A. thaliana*, *B. napus*, *S. lycopersicum*, *H. vulgare,* and many other species show that GAs play an important role in the fruit/pod setting and development [72,73,74,75]. Thus, in this paper, tissue and cellular localization of GA_3_ in the early and late stages of pod development was established. Insightful analyses have shown a moderate level of GA_3_ in pod walls of yellow lupine, which is associated with their role in young pod forming. In many plants, GAs promote early development of pod walls, which are extremely important in encapsulating the seeds and protecting them from, e.g., pests or pathogens [76]. In addition to this protective function, the photosynthetically active pod walls contribute assimilates and nutrients to fuel seed growth. The signals originating from the pod walls may also act to coordinate grain filling and regulate the reallocation of reserves from damaged seeds to those that have retained viability [76]. In *B. napus*, GAs coordinate the development of the pod walls and maintain it in a specific range, which is attributed to this phytohormone, where reduced GA level causes a decrease in the weight and size of the pods [73]. On the other hand, it is also well known that extra accumulation of GAs reduces the amount of seeds and often induces the development of parthenocarpic pods [72,73]. Therefore, during early pod development, it is extremely important to maintain an appropriate phytohormone balance. This was confirmed in *B. napus,* where correct GA-CK level allows proper development of pericarp [73]. During the late phase of pod development, the fluorescence signal of GA_3_ was undetectable in the pod walls, probably due to the fact that the growth of the pods has already finished. In turn, relationship between the observable content of GA_3_ and the low expression level of *LlDELLA1* in seed development may also confirm their important role in this process. 

## 4. Materials and Methods

### 4.1. Plant Material, Growing Conditions, and Phytohormone/Inhibitor Treatments

The seeds of yellow lupine (*Lupinus luteus* L.) epigonal cv. Taper were obtained from the Wiatrowo Plant Breeding Branch, Poznań Plant Breeding Station (Poznań, Poland). Before sowing, the seeds were treated with fungicide Vitavax 200FS solution (2.5 cm^3^/kg seeds, Chemtura AgroSolutions, Middlebury, CT, USA), and inoculated with Nitragina (3 g/kg seeds, BIOFOOD S.C, Wałcz, Poland) containing bacteria *Bradyrhizobium lupine*. The plants were cultivated on the 5^th^ soil class in the experimental field in Grubno in north-central Poland (53°20′31″N 18°28′12″E), thanks to cooperation with the Kuyavian-Pomeranian Agricultural Advisory Centre in Minikowo, Department in Przysiek (Przysiek, Poland), in accordance with the manufacturer’s agricultural recommendations [77]. The specimens, including leaf blades, petioles, stipules, stems, roots, flowers, and pods, was collected from plants with fully developed flowers (FDF) and fully developed pods (FDP). In addition, 10 phases of flower development (1F–10F) and 8 phases of pod development (1P–8P) were distinguished (Figure 6A). The appropriate material, with not less than 20 plants, was collected. Part of the flowers being in the 7th developmental phase (7F) was additionally treatment with gibberellic acid (GA_3_, 100 µM), GA biosynthesis inhibitor—chlorocholine chloride (CCC, 100 µM), methyl jasmonate (MeJA, 100 µM), indole-3-acetic acid (IAA, 100 µM), and 6-benzylaminopurine (BAP, 100 µM) in a 0.05% Tween 20 solution using a sprayer. The control flowers being at the identical developmental phase were treated in the same manner, but with a 0.05% Tween 20 solution only. After 3 h, the flowers were harvested. Depending on method then applied, generative organs were processed fresh or frozen in liquid nitrogen and stored at −80 °C. 

### 4.2. Molecular Cloning of LlDELLA1 cDNA

Frozen, fully developed flowers (~80 mg fresh weight) were homogenized in a sterile chilled mortar with a pestle. Total RNA was extracted using Isolate II RNA Plant Kit (Bioline, London, UK) according to the manufacturer’s protocol. Next, 1 μg RNA was reverse transcribed with a Transcriptor High Fidelity cDNA Synthesis Kit and oligo dT_(18)_ primers (Roche, Mannheim, Germany). Touchdown PCR (96 °C for 300 s; 40 cycles of 96 °C for 45 s, 65–60 °C for 45 s and 74 °C for 45 s; 74 °C for 420 s; cooling at 4 °C) was performed using the T3 Thermocycler (Biometra, Göttingen, Germany) with 1× buffer B, dNTP mix (0.2 mM), Mg^2+^ (3 mM), degenerate primers (1 μM) (Appendix A), Perpetual *Taq* DNA Polymerase^HOT START^ (1.25 U) (EURx, Warsaw, Poland), cDNA (0.5 μg) and deionized H_2_O up to a final volume of 50 μL. The partial cDNA was isolated, purified, cloned, and sequenced as described by Marciniak et al. [78]. The 3′ end of *LlDELLA1* cDNA was obtained using RACE-PCR technique (5′-3′ FirstChoice RLM-RACE Kit, SuperTaq-Plus Polymerase, Ambion, Inc., Austin, TX, USA) with designed primers (Appendix A). Due to difficulties arising from the experimental identification of the 5’ end of *LlDELLA1* cDNA, it was obtained based on sequences derived from a later RNA-Seq experiment deposited at NCBI in the Sequence Read Archive (SRA) database under accession number PRJNA285604 (BioProject, https://www.ncbi.nlm.nih.gov/bioproject/?term=PRJNA285604) and experiment accession number SRX1069734. Full-length *LlDELLA1* cDNA was deposited at the GenBank database (Acc no MN956900).

### 4.3. Bioinformatic Analyses 

The integrated FastPCR v.6.5.99 (http://primerdigital.com/fastpcr.html) tool was used for degenerate and RACE-PCR primers’ design, whereas Universal Probe Library Assay Design Center (http://www.roche-applied-science.com/sis/rtpcr/upl) was used for design of qPCR specific primers and probes. The identified sequence was analyzed using BLAST (https://blast.ncbi.nlm.nih.gov/Blast.cgi) and ExPASY (http://www.expasy.org/), including Translate (https://web.expasy.org/translate/) and ProtParam (https://web.expasy.org/protparam/) tools. Alignment and phylogenetic reconstructions were performed using the function “build” of Environment for Tree Exploration (ETE3) v3.1.1 [79] as implemented on the GenomeNet (https://www.genome.jp/tools/ete/). Maximum likelihood tree was inferred using PhyML v20160115 ran with model and parameters: --alpha e -f m --pinv e --nclasses 4 -o tlr --bootstrap -2 [80]. Branch supports are the Chi^2^-based parametric values returned by the approximate likelihood ratio test. To identify motifs within the DELLAs in different plant species, the MEME motif search tool v.5.1.0 was used (http://meme-suite.org/) [81] with default settings, except the maximum number of motifs to be found was set at 30. Multiple alignments of DELLAs from *L. luteus* and *A. thaliana* were made using the DiAlign program (Genomatix) (http://www.genomatix.de/cgi-bin/dialign/dialign.pl) with default parameters. The ProtComp v.9.0 program was used to predict the sub-cellular localization of the LlDELLA1 and all AtDELLA (http://www.softberry.com). The tertiary structure of the LlDELLA1 was constructed using protein structure prediction Robetta server (http://new.robetta.org/) [82] and protein fold recognition phyre^2^ server (http://www.sbg.bio.ic.ac.uk/~phyre2/html/page.cgi?id = index) [83]. The results were visualized using the VMD 1.9.3 program [84]. The analysis of secondary structure was performed using the STRIDE web server (http://webclu.bio.wzw.tum.de/stride/) [85].

### 4.4. Expression Analysis

Expression profile of *LlDELLA1* in various tissues and in response to hormone treatments was analyzed via quantitative real-time PCR. The cDNAs were obtained in the same manner as previously described for the cloning of *LlDELLA1*. qPCR containing 0.2 μM gene-specific primers (Appendix A), 0.05 μM Universal Probe Library (UPL) hydrolysis probes (Roche) (Appendix A), 0.1 μg of cDNA and 1 × LightCycler TaqMan Master Mix (LightCycler TaqMan Master Kit, Roche) was performed in 20 μL glass capillaries using a LightCycler 2.0 Carousel-Based System (Roche, Mannheim, Germany). cDNA-free negative controls were included. As a reference endogenous control for normalization purposes, the actin gene (*LlACT*) was chosen [78,86,87]. The reactions were carried out as follows: 600 s at 96 °C; 45 cycles of 10 s at 96 °C, 15 s at 58 °C, 1 s at 72 °C; and 30 s at 40 °C. Absolute quantification was calculated using the standard curves from serial dilutions of cDNAs templates for both the studied and reference genes. Relative gene expression, which presents the data of the *LlDELLA1* gene relative to calibrator and internal control gene, was determined using the 2 (-DeltaDeltaC(T)) method [88].

### 4.5. Histological Studies and GA_3_ Immunolocalization

The appropriate tissue fragments were fixed in 4% paraformaldehyde (*w*/*v*), 0.2% glutaraldehyde (*v*/*v*) and 3% N-ethyl-N′-(3-dimethylaminopropyl)carbodiimide hydrochloride (EDAC) (*w*/*v*) (Sigma-Aldrich, St. Louis, MO, USA) prepared in 1× phosphate-buffered saline buffer (1× PBS, pH 7.2) for 12 h at 4 °C overnight. The samples were washed in 1× PBS (pH 7.2), dehydrated in increasing ethanol concentrations (30%, 50%, 70%, 90%, 100%) (*v/v*), supersaturated, and embedded in BMM resin (butyl methacrylate, methyl methacrylate, 0.5% (*w/v*) benzoin ethyl ether, 10 mM dithiothreitol) (Fluka, Buchs, Switzerland) at −20 °C using UV light for polymerization. Semithin sections (1.5 μm) were cut on an Ultracut microtome (Reichert-Jung, Germany), placed on glass slides covered with Biobond (BBInternational, Cardiff, UK) and used for histochemical staining and immunolocalization studies. 

Preparations for histological examination were stained with 0.05% toluidine blue (Sigma-Aldrich, Saint Louis, MO, USA). Sections were observed in the LM Zeiss Axioplan (Carl Zeiss, Oberkochen, Germany) microscope equipped with a ProGres C3 digital camera.

The sections on slides for immunofluorescence studies were blocked in BlockAid TM Blocking Solution (Thermo Fisher Scientific, Waltham, MA, USA) according to the manufacturer’s instruction. Then, sections were incubated o/n at 4 °C with the rabbit polyclonal primary antibody anti-GA_3_ (Abbexa Ltd., Cambridge, UK) diluted 1:50 in 1% bovine serum albumin (BSA) in 1 × PBS (pH 7.2). Next, a DyLight Alexa 488 conjugated IgG diluted 1:250 in PBS buffer for 2 h at 37 °C was served as the secondary antibody (Agrisera, Vännäs, Sweden). Negative control reaction, required for validation of the immunohistochemical findings, was carried out by omitting the incubation with the primary antibody, and showed no labeling (Appendix A). The samples were observed in a Leica DMI4000B inverted microscope using the BP365, FT395, and LP397 filters.

### 4.6. Statistical Analysis

All presented data are the results of three separate samples (biological replications) with three repetitions of each (technical replications) (*n* = 9) and presented as mean ± standard error (SE). Statistical analysis was performed using one-way Anova followed by post-hoc Tukey’s HSD test, with differences accepted at *p* > 0.05. All analyses were performed using R version 3.5.3.

## Figures and Tables

**Figure 1 ijms-21-01815-f001:**
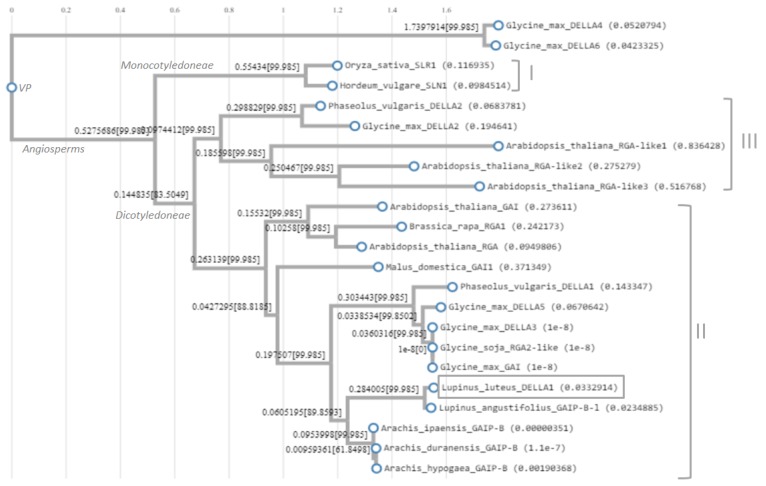
Maximum likelihood phylogenetic tree of 23 DELLAs derived from different plant species. Alignment and phylogenetic reconstructions were performed using Environment for Tree Exploration3 (ETE3) v3.1.1 program, as implemented on the GenomeNet. ML tree was inferred using PhyML v20160115 ran with model and parameters: --alpha e -f m --pinv e --nclasses 4 -o tlr --bootstrap -2. Branch supports are the Chi^2^-based parametric values returned by the approximate likelihood ratio test. *Lupinus angustifolius* GAIP-B-like (XP_019460121.1); *Arachis hypogaea* GAIP-B (XP_025691574.1); *A. duranensis* GAIP-B (XP_015957883.1); *A. ipaensis* GAIP-B (XP_016191184.1); *Glycine max* GAI (ALR99819.1); *Glycine soja* RGA2-like (XP_028232592.1); *Malus domestica* DELLA (ADW85805.1); *Arabidopsis thaliana* GAI (CAA75492.1); *A. thaliana* RGA (CAA72177.1); *A. thaliana* RGA-like1 (NP_176809.1); *A. thaliana* RGA-like2 (NP_186995.1); *A. thaliana* RGA-like3 (NP_197251.1); *Oryza sativa* DELLA (BAE96289.1); *Hordeum vulgare* SLN1 (AAL66734.1); *Brassica rapa* DELLA (AAX33297.1); *G. max* GAI/DELLA2 (XP_003552980.1); *G. max* GAI1/DELLA3 (NP_001240948.1); *G. max* SLR1/DELLA4 (XP_003528281.1); *G. max* GAI1/DELLA5 (XP_003531153.1); *G. max* DWARF8/DELLA6 (XP_003524001.1); *Phaseolus vulgaris* DELLA1 (BAF62636.1); *P. vulgaris* DELLA2 (BAF62637.1). VP—vascular plants.

**Figure 2 ijms-21-01815-f002:**
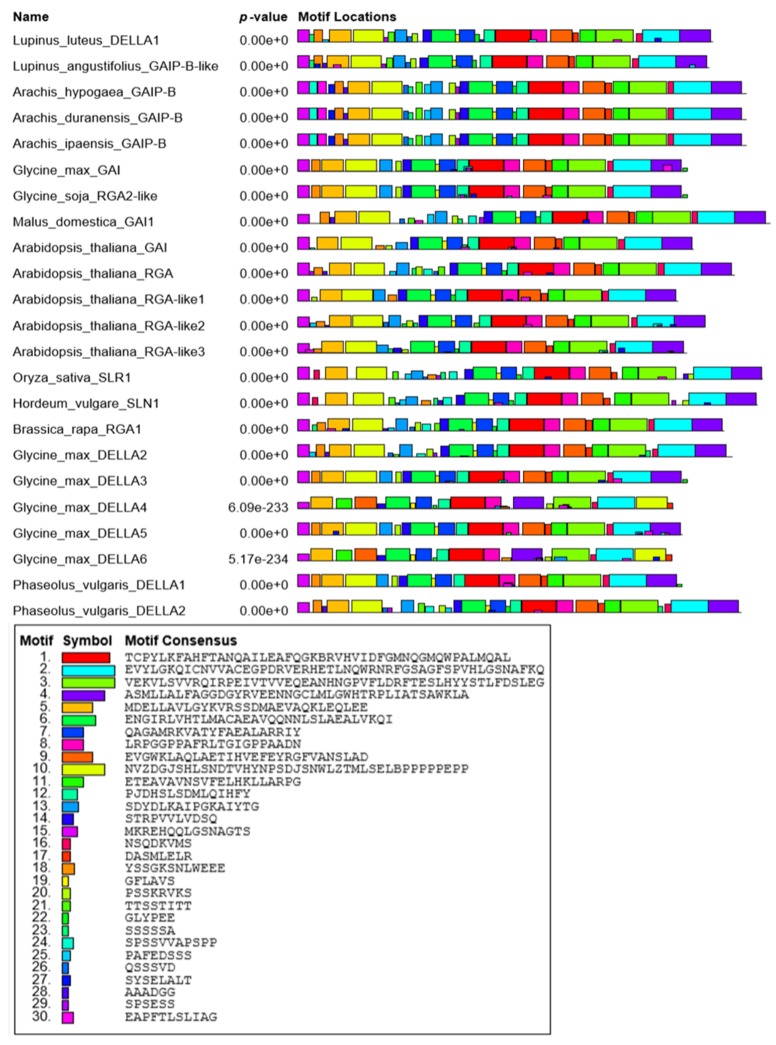
This diagram shows the location of motif sites in 23 DELLAs from different plant species. Each block shows the position and strength of a motif site. The height of a block gives an indication of the significance of the site, as taller blocks are more significant. The height is calculated to be proportional to the negative logarithm of the *p*-value of the site, truncated at the height for a *p*-value of 1 × 10^-10^.

**Figure 3 ijms-21-01815-f003:**
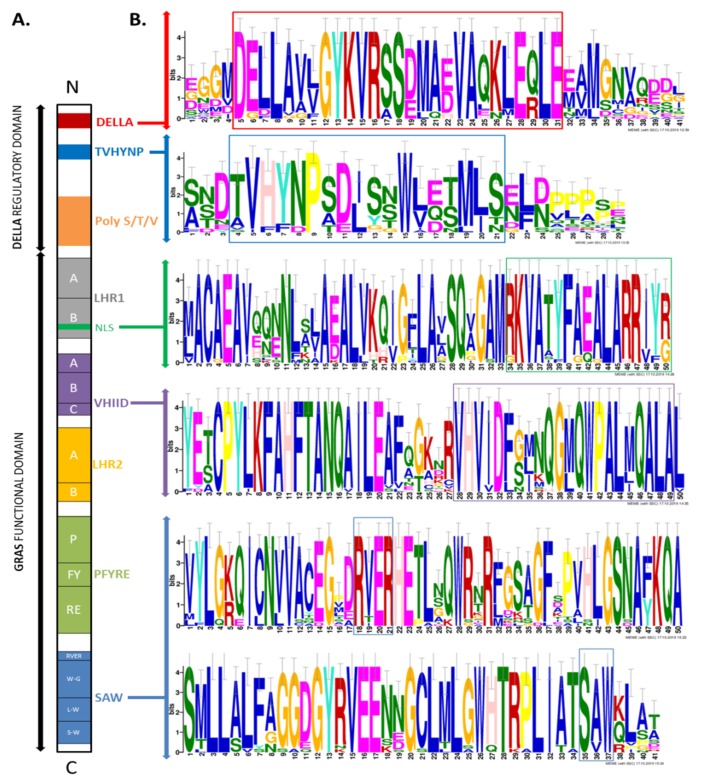
(**A**) Domain structure of a typical DELLA protein; (**B**) The motifs identified in the 23 DELLAs in different plant species. The overall height of each stack indicates the sequence conservation at that position, whereas the height of symbols within each stack reflects the relative frequency of the corresponding amino acid (MEME motif search tool).

**Figure 4 ijms-21-01815-f004:**
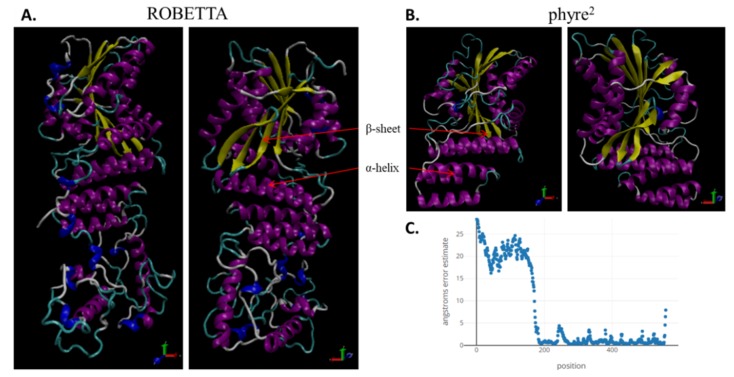
3-D model of LlDELLA1 constructed by **(A)** ROBETTA and **(B)** phyre^2^ protein modeling servers visualized using the VMD program. The model predicted by ROBETTA has 17 α-helixes, β-sheet with 9 strands and covers the entire amino acid sequence (**A**). This model has 63% confidence level, which might be a result of lack of appropriate template from Protein Data Bank (PDB) in the first 180 aa (**C**). The phyre^2^ predicted 3D structure between 186 aa and 554 aa. This model has 11 α-helixes, β-sheet with 9 strands and has 100% confidence; it covers 66% of the sequence (**B**).

**Figure 5 ijms-21-01815-f005:**
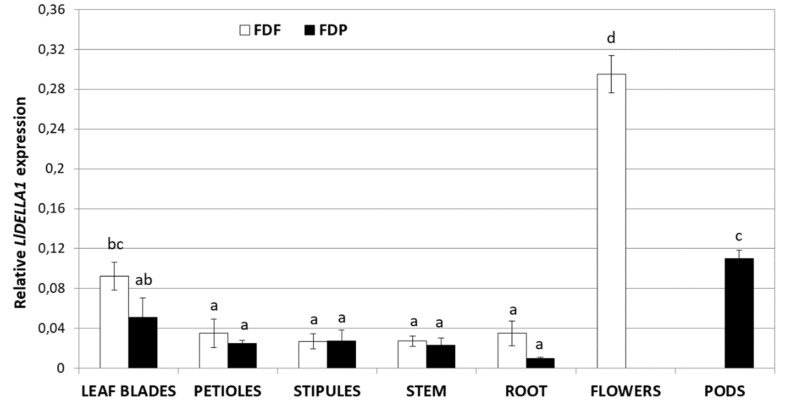
Expression profile of *LlDELLA1* in vegetative (leaf blades, petioles, stipules, stem, root) and generative (flowers, pods) tissues of yellow lupine during generative development—fully developed flowers (FDF) and fully developed pods (FDP). The average value with standard errors (SE) of the three repetitions containing three replicates (*n* = 9) was used to draw the figure. Letters represent statistically significant differences at *p* < 0.05 (one-way ANOVA followed by Tukey’s Honest Significant Difference test).

**Figure 6 ijms-21-01815-f006:**
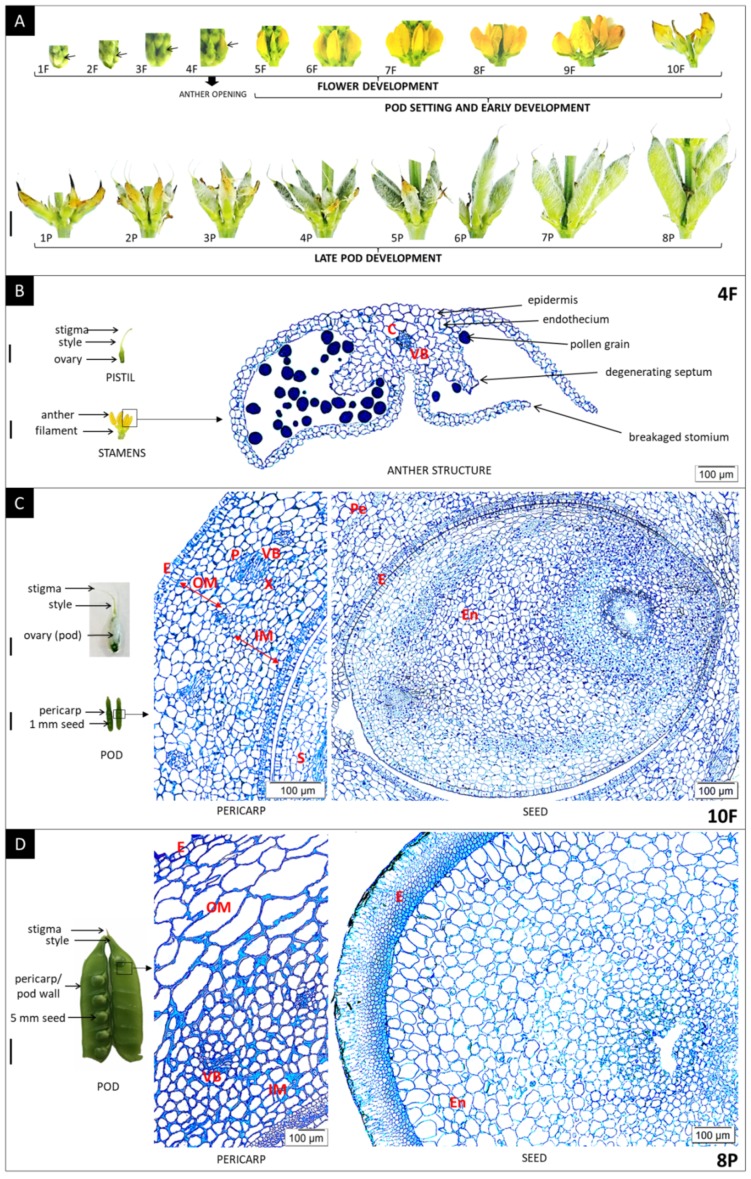
(**A**) Individual stages of flower (1F–10F) and pod development (5F–10F—pod setting and early development; 1P–8P—late pod development) in yellow lupine. After dehiscence program, anther opening occurred in the ~4F phase, followed by pollination and fertilization. (**B,C,D**) The anatomical structure of different generative organs (cross-sections). The anthers were collected from plants in the fourth stage of flower development (4F) (**B**). The pod wall/pericarp and seed were collected from plants in the tenth stage of flower development (10F) (**C**) and from plants in the eighth stage of pod development (8P) (**D**). Sections were stained with toluidine blue. C—connective, VB—vascular bundle, P—phloem, X—xylem, E—epidermis, OM—outer mesocarp, IM—inner mesocarp, Pe—pericarp, En—endosperm. Scale bars = 1 cm (A,D); 0.2 cm (B, each); 0.5 cm (C, each).

**Figure 7 ijms-21-01815-f007:**
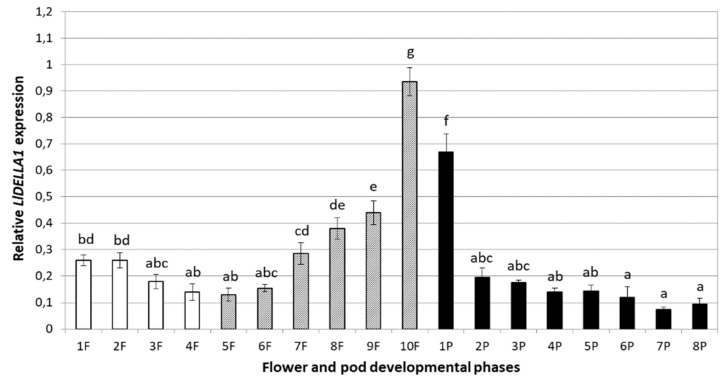
Expression profile of *LlDELLA1* in different stages of generative organ development—flowers (1F–10F), early development of pods (5F–10F, marked in gray), and late development of pods (1P–8P, marked in black) in yellow lupine. The average value with ± SE of the three repetitions containing three replicates (*n* = 9) was used to draw the figure. The letters represent statistically significant differences at *p* < 0.05 (one-way ANOVA, followed by Tukey’s HSD test).

**Figure 8 ijms-21-01815-f008:**
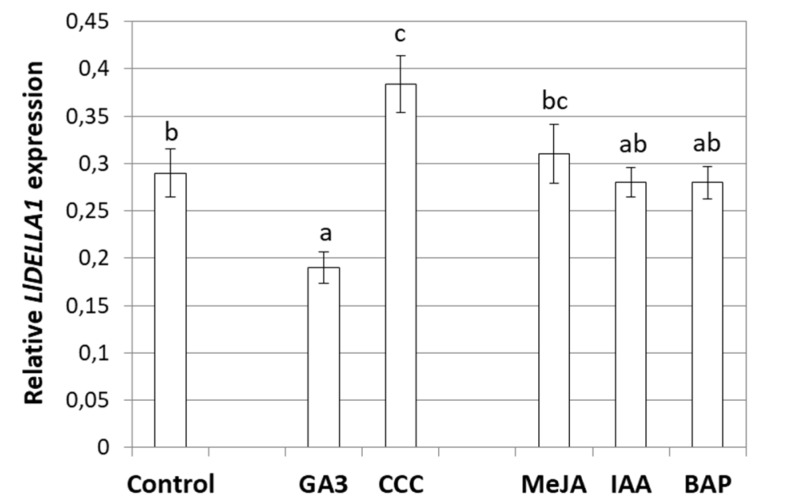
Expression profile of *LlDELLA1* in yellow lupine flowers after different compound treatments. GA_3_—gibberellic acid, CCC—chlorocholine chloride, MeJA—methyl jasmonate, IAA—indole-3-acetic acid, BAP—6-benzylaminopurine. The transcriptional activity of *LlDELLA1* was measured in three independent biological replicates and three technical replicates (*n* = 9); ± SE is marked on the bars. Letters represent statistically significant differences at *p* < 0.05 (one-way ANOVA, followed by Tukey’s HSD test).

**Figure 9 ijms-21-01815-f009:**
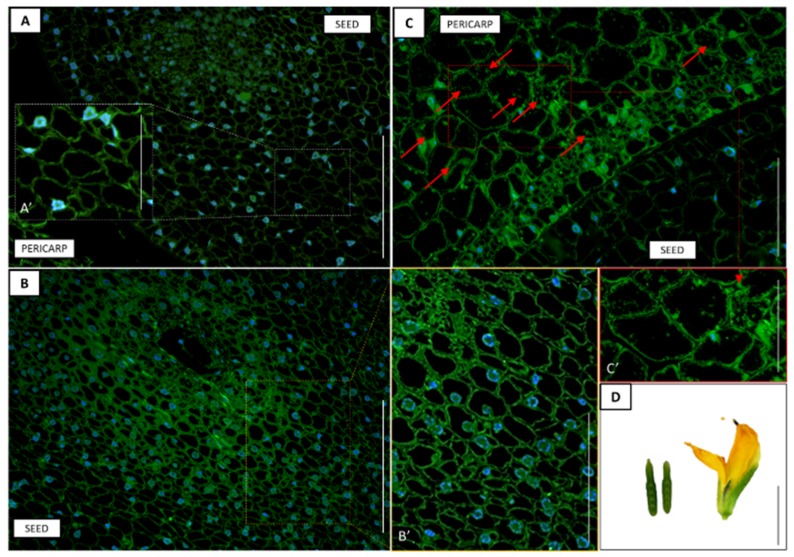
Tissue and cellular localization of gibberellic acid (GA_3_) (**A**–**C**) in the early phase (10F; **D**) of pericarp and seed development in yellow lupine. Moderately low but visible GA_3_ level was observed in the pericarp (**C,C’**). In seeds, the phytohormone molecules were undetectable (**A,A’,B,B’**). The subfigures A', B' and C' are an enlargement of A, B and C, respectively. Red arrows indicate the localization of GA_3_. DAPI was used for nuclei staining. Scale bars: 100 µm (A,B), 50 µm (B’,C), 30 µm (C’), 25 µm (A’), 1 cm (D).

**Figure 10 ijms-21-01815-f010:**
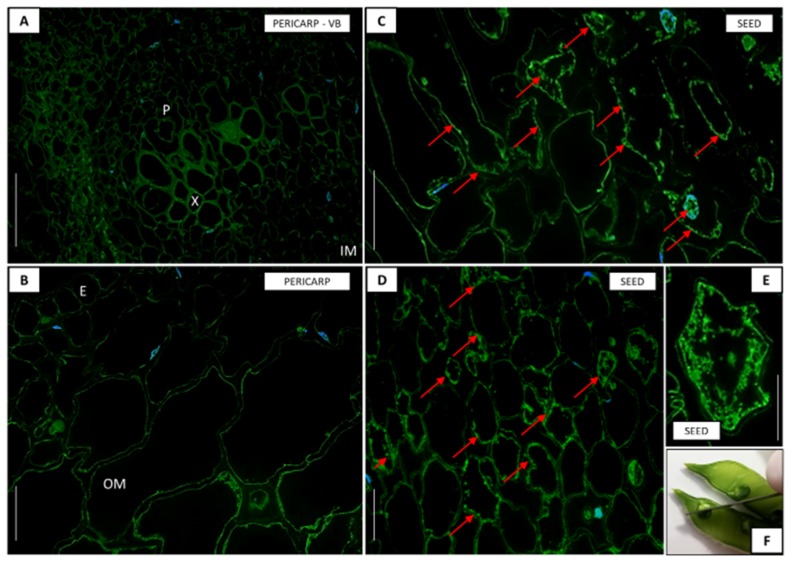
Tissue and cellular localization of gibberellic acid (GA_3_) (**A**–**E**) in the late phase (8P; **F**) of pericarp and seed development in yellow lupine. A higher level of GA_3_ was detected in the seeds (**C,D,E**) than in the pericarp (**A,B**). The green fluorescence signal was mainly observed near the walls of the seed cells (**C,D**) and only in some cells were the signal distributed throughout the cytoplasm (**E**). In pericarp, the presence of GA_3_ was not observable for epidermis (E), outer mesocarp (OM), and inner mesocarp (IM) cells, as well as conductive bundles (P—phloem, X—xylem) (**A,B**). Red arrows indicate the localization of GA_3_. DAPI was used for nuclei staining. Scale bars: 50 µm (A,B), 30 µm (C,D), 20 µm (E).

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
