# Peer review of "Gibberellin Signaling Repressor LlDELLA1 Controls the Flower and Pod Development of Yellow Lupine (Lupinus luteus L.)"

_ijms, 2020, doi:10.3390/ijms21051815_

Round 1

Reviewer 1 Report

The manuscript describes a work about a key repressor of gibberellin signaling (DELLA) in Lupinus luteus, a legume crop which struggles to spread because a number of issues including environmental adaptability and diseases susceptibility.
The work is well presented and many experimental data with great details are given (there are as many as 12 figure and 11 supplementary materials).

The manuscript is generally well written and the text is well cared for. I suggest to include at the end of the manuscript the full list of used abbreviations.

Introduction:
row 34. Is the problem circumscribed to Poland or is a general problem?
row 43. Please delete " Through physical interactions"
row 71. Have been explored in other lupin species? To the best of my knowledge it have not. Can the Authors rephrase the incipit of the sentence?

Materiale and methods:
row 90. Rather than "The research material..." another term should be used (Samples? Specimens?, etc)
row 150. Please specify details about biological replicas, too. Always 80 mg fresh weight for all tissues/sample? I believe the detail is reported in the legend of figure 10, but it is better to specify the detail in M&M.

Results:
fig 5B. Did the Authors used Weblogo application (at Berkeley.edu) to analyse amino acid frequencies, or with other program ?
row 245. ‘Φ’ represents a non-polar residue. This should be specified here (now is in the legend of one supplementary figure.
row 256. Please comment the differences observed using the two tools. Which is the plus of having used two different algorithms?

The Author should state if the conclusions stemmed by their work may be extended to other Lupin species ((are known any functional differences among the three main cultivated lupin species) or other legume crops ?

Author Response

Response to Reviewer 1

Dear Reviewer,

We want to thank you very much for giving us the opportunity to improve our manuscript entitled “Gibberellin signaling repressor LlDELLA1 controls the flower and pod development of yellow lupine (Lupinus luteus L.)” to be considered for publication in the International Journal of Molecular Sciences. Each of the suggestions and comments was extremely important to us and has definitely improved the quality of our manuscript. We have made an effort to respond in detail to all suggestions. We believe that the revised manuscript will meet your expectations. Below are our answers to the comments and suggestions.

  1. I suggest to include at the end of the manuscript the full list of used abbreviations.

The full list of used abbreviations was added at the end of the manuscript (Please see the lines 607-620).

Introduction:

  1. row 34. Is the problem circumscribed to Poland or is a general problem?

Thank you for your question that made us think deeply. Premature and extensive generative organ abortion is general lupine problem which all over the globe is connected with the influence of different endogenous and exogenous factors on this process. For example, Australian lupine production is reduced dramatically due to generative organ abscission resulting mainly from drought stress conditions (the data from FAO). In Poland this problem has various causes, such as insufficient, as quantitative as well as qualitative pollination of flowers, different availability of assimilates and mineral substances or diverse distribution and activity of plant hormones. In order not to mislead the future reader, we have removed the error in the manuscript "in Poland" (L 35).

  1. row 43. Please delete " Through physical interactions"

The sentence has been corrected as suggested (L 45).

  1. row 71. Have been explored in other lupin species? To the best of my knowledge it have not. Can the Authors rephrase the incipit of the sentence?

Yes, you have right; this has not been studied in other lupine species. The sentence has been changed as suggested (L 88-89).

Material and methods:

  1. row 90. Rather than "The research material..." another term should be used (Samples? Specimens?, etc)

We changed “research material” to “specimens” (Please see the L 510).

  1. row 150. Please specify details about biological replicas, too. Always 80 mg fresh weight for all tissues/sample? I believe the detail is reported in the legend of figure 10, but it is better to specify the detail in M&M.

Thank you for your suggestion. The information about biological replicates was included in “Material and methods” section in the line 521, and about technical replicates in the line 577. Now, we ordered this information in newly created section “Statistical analysis” (Please see the lines 602-606).

When isolating total RNA using Total RNA Isolate II RNA Plant Kit we always use about 80 g of fresh tissue in each case because this amount of plant material does not clog up homogenizing and binding columns, which we have experimentally tested. The quantity and quality of RNA obtained in this way is very good and is suitable for further applications of molecular biology. Additionally, in the manuscript we have added “~ 80 g” (L 523).

Results:

  1. fig 5B. Did the Authors used Weblogo application (at Berkeley.edu) to analyse amino acid frequencies, or with other program?

Thank you for your question. To identify motifs within the 23 DELLAs in different plant species, the MEME motif search tool v.5.1.0 was used (http://meme-suite.org/) with default settings, except the maximum number of motifs to be found was set at 30. A minor correction was made in the “Materials and methods” section (L 552-553), and this information was also added in the description of the fig. 3 (please see the line 160).

  1. row 245. ‘Φ’ represents a non-polar residue. This should be specified here (now is in the legend of one supplementary figure.

“‘Φ’ represents a non-polar residue, and ‘x’ can represent any residue” - this phrase was added in the manuscript between the lines 166-167.

  1. row 256. Please comment the differences observed using the two tools. Which is the plus of having used two different algorithms?

The use of two independent approaches in protein structure prediction allows for more reliable results. Initially, we used the Phyre2 program, whose algorithms are based on comparative modeling of protein. Using this tool, we obtained a model consisting only of a portion of the protein most similar to other proteins having experimentally known structures, mostly GRAS family members. The most diverse sequence fragments were omitted due to the lack of homological templates in PDB. Therefore, a different approach was used to obtain the entire protein model. I-TASSER program was used, which does not take into account sequence homology, but uses combination of ab initio folding and threading methods. The results obtained by this method were not satisfactory, and the protein model did not show statistical significance. So the program that combines homology modeling and ab initio fragment assembly was used. This allowed to obtain a model generated from the entire sequence, which showed high statistical significance in the conserved sequences, while worse predictions in highly variable sequences.

  1. The Author should state if the conclusions stemmed by their work may be extended to other Lupin species (are known any functional differences among the three main cultivated lupin species) or other legume crops?

We've included a brief commentary on this topic in the introduction section (please see the lines 98-104).

The results of our research conducted for years on various species and varieties of lupins (Lupinus luteus, Taper and Mister; L. albus, Butan and Boros; L. angustifolius, Kadryl and Sonet) indicate that the genes associated with GA metabolism and signaling pathways show a high degree of identity and similarity. Therefore, there are strong premises that the knowledge obtained for one variety or even a species can be transferred to another. The differences most likely appear at the level of proteins and how they are regulated.

Reviewer 2 Report

In this manuscript, a DELLA cDNA is identified in Lupinus luteus and, among other properties, its expression during flower and pod development is presented. Although the experiments are well conducted, I have some doubts about if the manuscript is adequate for publication in International Journal of Molecular Sciences, or if a more specific journal is more appropriate.

I have some major concerns that could improve the manuscript:

  • The Introduction could be improved.
  • I find the Figure 1 dispensable in the paper.
  • The Figure 2 should be passed to supplementary material or as pannel in Figure 8.
  • I cannot find if the authors performed a PCR to amplify and sequence the whole cDNA in order to demonstrate the mixing among several genes. In the Introduction, single copy genes are presented for several species except for Arabidopsis. However, in the Figure 3 several copies are presented for DELLA in the legumes included in the tree (Glycine and Phaseolus). Therefore, I am not sure if Lupinus luteus has a unique DELLA gene or several as the other legumes.
  • The nomenclature in Figure 8 (4th FD, 8th FD, 8th PD) is different to Figure 2 and Figure 9.
  • The authors should consider if Figure 8 is important for the understanding of the manuscript.
  • The statistical analysis could be interesting at least in Figure 10.
  • It is not clear why the authors tested the compounds effect in Figure 10 only after 3 hours of treatment. Perhaps, the effect of CCC and GA3 can be clearer with other treatment-times.

Author Response

Response to Reviewer 2

Dear Reviewer,

We want to thank you very much for giving us the opportunity to improve our manuscript entitled “Gibberellin signaling repressor LlDELLA1 controls the flower and pod development of yellow lupine (Lupinus luteus L.)” to be considered for publication in the International Journal of Molecular Sciences. Each of the suggestions and comments was extremely important to us and has definitely improved the quality of our manuscript. We have made an effort to respond in detail to all suggestions. We believe that the revised manuscript will meet your expectations. Below are our answers to the comments and suggestions.

  1. The Introduction could be improved.

The introduction section has been improved. Additional information has been appended that will better introduce the future reader to the research topic (Please see the lines 58-72; 79-88; 98-104).

  1. I find the Figure 1 dispensable in the paper.

Figure 1 has been removed from the paper.

  1. The Figure 2 should be passed to supplementary material or as pannel in Figure 8.

Thank you for this suggestion. Figure 2 was combined with Figure 8. Currently it is Figure 6.

  1. I cannot find if the authors performed a PCR to amplify and sequence the whole cDNA in order to demonstrate the mixing among several genes. In the Introduction, single copy genes are presented for several species except for Arabidopsis. However, in the Figure 3 several copies are presented for DELLA in the legumes included in the tree (Glycine and Phaseolus). Therefore, I am not sure if Lupinus luteus has a unique DELLA gene or several as the other legumes.

Thank you for your insight. A single cDNA of the yellow lupine LlDELLA1 gene was identified in this study. For this purpose, we used mixed methods: RT-PCR reactions with degenerate primers and 3’ RACE-PCR, while the 5 'end fragment comes from transcriptomic data. We explained it in the materials and methods section (please see the lines 522-539). We would like to point out that degenerate primers were designed based on cDNA sequences derived from closely related species (Glycine max, Phasoleus vulgaris, Malus domestica) to yellow lupine (these 3 sequences were found in the BlastN database). In further analyzes of the LlDELLA1, we compared it with different protein sequences come from various species, both closely related and those most commonly studied (e.g. Arabidopsis thaliana). We suspect that the information in the introduction section may have been misleading. Now they have been improved (please see the lines 58-72). At this point, we suspect that yellow lupine may have several DELLA genes, but this requires confirmation and further research. Moreover, the LlDELLA1 identified by us in this work plays a key role in the development of generative organs.

  1. The nomenclature in Figure 8 (4th FD, 8th FD, 8th PD) is different to Figure 2 and Figure 9.

According to the valuable reviewer's suggestion, the entire nomenclature was unified on all Figures.

  1. The authors should consider if Figure 8 is important for the understanding of the manuscript.

At present, Fig. 8 has been combined with Fig. 2 as described above. In our opinion, an improved Fig. 6 is important for the understanding of the manuscript, at least due to the possibility of better interpretation of immunohistochemical results. In yellow lupine this is a pioneering study on the structure of flowers and pods. We have selected individual stages of generative organ development and we will refer to them in our next papers. We believe that this will also be useful for other researchers to study similar physiological processes at the molecular level. We hope you will be convinced by our arguments.

  1. The statistical analysis could be interesting at least in Figure 10.

The complex statistical analysis was included in all figures. Additionally, “Statistical analysis” section was created (Please see the line 602-606).

  1. It is not clear why the authors tested the compounds effect in Figure 10 only after 3 hours of treatment. Perhaps, the effect of CCC and GA3 can be clearer with other treatment-times.

The results of many papers, involving the treatment of plants with various hormones allow us to state that the first response at the molecular level (changes in transcriptional activity) appear even after a few minutes after compound application (usually 1-2 hours). Therefore, three hours seem to be the optimal time to measure the amount of LlDELLA transcripts after hormone application. During the research, the material was also collected 24 hours after compound application, but the results obtained were similar to those obtained after 3 hours, therefore they were not included. The main purpose of this task was to prove that the identified LlDELLA1 works in a GA-dependent manner but not the influence of treatment-time on LlDELLA1 expression.

  1. Moore, T.C. Biochemistry and Physiology of Plant Hormones.
  2. El-Sharkawy, I.; Sherif, S.; Abdulla, M.; Jayasankar, S. Plum fruit development occurs via gibberellin–sensitive and –insensitive DELLA repressors. PLoS ONE 2017, 12, e0169440.
  3. Zentella, R.; Zhang, Z.-L.; Park, M.; Thomas, S.G.; Endo, A.; Murase, K.; Fleet, C.M.; Jikumaru, Y.; Nambara, E.; Kamiya, Y.; Sun, T.P. Global analysis of DELLA direct targets in early gibberellin signaling in Arabidopsis. Plant Cell 2007, 19, 3037-3057.

Round 2

Reviewer 2 Report

The authors have addressed my main concerns in an adequate way and, therefore, I consider that the revised manuscript has been improved. Mainly, the introduction is now more complete and it made the manuscript easier to follow. In relation with the hormone treatment experiment, the authors could consider to include a sentence indicating that the same result was obtained after 24 hours treatment. Especially when the induction/repression on gene expression at 3 hours treatment is in the 25-35% range.